# Caffeine-Containing, Adaptogenic-Rich Drink Modulates the Effects of Caffeine on Mental Performance and Cognitive Parameters: A Double-Blinded, Placebo-Controlled, Randomized Trial

**DOI:** 10.3390/nu12071922

**Published:** 2020-06-29

**Authors:** Ali Boolani, Daniel T. Fuller, Sumona Mondal, Tyler Wilkinson, Costel C. Darie, Eric Gumpricht

**Affiliations:** 1Department of Physical Therapy, Clarkson University, Potsdam, NY 13699, USA; 2Department of Mathematics, Clarkson University, Potsdam, NY 13699, USA; fullerdt@clarkson.edu (D.T.F.); smondal@clarkson.edu (S.M.); 3Department of Chemistry and Biomolecular Science, Clarkson University, Potsdam, NY 13699, USA; wilkintf@clarkson.edu (T.W.); cdarie@clarkson.edu (C.C.D.); 4Isagenix International, LLC, Gilbert, AZ 85297, USA; Eric.gumpricht@isagenixcorp.com

**Keywords:** adaptogens, caffeine, cognition, energy, fatigue, mood

## Abstract

Using a placebo-controlled, double-blinded, within-participants, randomized, cross-over design, we examined the neurocognitive effects of a: (a) caffeine-containing, adaptogenic herbal-rich natural energy shot (e+ shot), (b) a matched caffeine-containing shot (caffeine), and, (c) a placebo. Participants (*n* = 30) were low consumers of caffeine without elevated feelings of energy. Before and three times after beverage consumption, a 27-min battery was used to assess motivation to perform cognitive tasks, mood, attention ((serial subtractions of 3 (SS3) and 7 (SS7), the continuous performance task (CPT), and the rapid visual input processing tasks)), heart rate (HR), blood pressure (BP), and motor coordination (nine-hole peg test) with a 10-min break between each post-consumption battery. The procedure was repeated for each beverage for each participant at least 48 h apart and within 30 min the same time of day using a random group assignment with blinding of researchers and subjects. To evaluate for changes in outcomes, a Treatment × Time analysis of covariance controlling for hours of prior night’s sleep was used. Analysis of all outcomes and all treatment comparisons indicated that compared to placebo, both e+ shot ($\bar{\Delta }\;$= 2.60; η^2^ = 0.098) and caffeine ($\bar{\Delta }\;$= 5.30, η^2^ = 0.098) increased systolic BP 30 min post consumption (still within normal healthy ranges). The caffeine beverage also led to an improvement in most cognitive measures and moods 30-min post-consumption with improvements tapering at 69 and 108 min, while e+ shot noted more steady improvements with no significant differences between beverages on most cognitive and mood measures at 69 and 108 min. However, compared to caffeine, e+shots noted a significant decrease in reaction time at 108 min, while caffeine noted a small change in the opposite direction. No side-effects were reported by any intervention. These results suggest that the specific blend of adaptogens in e+ shot may modulate the neurocognitive effects of caffeine on mood, and cognition.

## 1. Background

Caffeine is the most commonly consumed psychoactive substance in the world [1] and a recent evaluation of 2011–2012 National Health and Nutrition Examination Surveys (NHANES) data estimated its intake for the average American adult at 173 milligrams (mg) per day (equivalent to one strong espresso coffee) [2]. The intake of dietary caffeine is primarily derived through consumption of beverages including coffee, tea, and cola products. There has been considerable research into its metabolic, physiologic, and neurophysiologic effects [3,4,5]. For example, acute caffeine consumption promotes psychostimulatory benefits including enhanced cognitive task performance and sustained attention [6], and improvements in working memory [7], visual reaction [8], logical reasoning [9], and manual dexterity [10]. However, numerous side effects have also been reported, including decreased calmness and increased anxiety [11], elevated blood pressure [11], withdrawal effects of headaches, drowsiness, and fatigue [12], sleep interference [13] and an increased risk for osteoporosis [14]. It has been suggested that the factors motivating regular caffeine consumption appear to be “withdrawal relief” [15].

In concordance with its demonstrated and perceived benefits, there has been a significant increase in the popularity of caffeine-containing energy beverages [15]. In most commercially-available energy drinks the primary bioactive is (synthetic) caffeine, followed by other ingredients, such as taurine, B vitamins (riboflavin, pyridoxine, and nicotinamide), and various herbal derivatives or synthetic ingredients present in lesser amounts [16]. A myriad of energy beverages are marketed with varying caffeine content (50–505 mg per can or bottle) and concentrations (0.09–6.02 mg per mL) [17]. Research into the metabolic and physiological responses to energy drink consumption has reported several positive beneficial effects akin to those observed with caffeine alone. Some of these benefits include: increased information processing, vigilance, memory, and verbal reasoning [11,17,18], improved mood characterized by increased alertness, vigor, and decreased fatigue [11,19], and decreased heart rate [20]. Conversely there are numerous acute side effects including increased anxiety [11,21], elevated blood pressure [11,22], and acute cardiac events [23]. It is unclear how variables such as ingredient composition and/or interactions, level of usage, genetics (i.e., polymorphisms in cytochrome P450 1A2 (CYP1A2), and demographic status of the individual (i.e., teenagers and young adults, gender) factor into these adverse responses. Consequently, some food manufacturers are seeking novel formulations to provide the consumer with the perceived benefits of caffeine-containing energy beverages while minimizing potential side effects.

In this context, novel and unique functional ingredients have been introduced into caffeine-containing energy products. One class of ingredients recently formulated into energy shots are adaptogens. Adaptogenic plants, although extensively studied by Russian researchers in the 1940s and onward [24,25], have only relatively recently begun attracting greater scientific exploration. Most studied among these plants are *Rhodiola rosea*, *Withania somnifera (ashwagandha)*, *Schisandra chinensis,* and *Eleutherococcus senticosus*. As a class of compounds, adaptogens have been shown to modify cellular stress responses, exert anti-fatigue activity, and improve mental work capacity [26,27]. Although the precise mechanisms by which adaptogens may influence these cellular responses are poorly characterized their effects are most likely non-specific in nature via modification of a wide array of cellular signaling pathways [28].

The primary purpose of the current study was to compare the acute effects a naturally-sourced caffeine and adaptogenic-rich energy shot, (e+ shot), to both placebo and a caffeinated, dose-matched active comparator on a variety of mental performance and cognitive parameters in young adults.

## 2. Methodology

### 2.1. Design and Study Products

This placebo-controlled, double-blinded, within-participants, randomized cross-over study examined the effects of three 60 mL interventions: a placebo, an active comparator (caffeine), and e+™ energy shot (e+ shot, Isagenix International, LLC, Gilbert, AZ, USA), a naturally-sourced caffeine and adaptogen-containing energy shot. To ensure effective blinding, none of the scientists conducting the study or analyzing the data were aware of the treatment assignments. Only one of the co-authors (EG) who coded the treatments as “A”, “B”, or “C” was aware of treatment allocation. Subsequently, the primary author (AB) randomly assigned the delivery and order of beverage treatments to the subjects. Participants received the treatment in an unmarked white container with a black top. All treatments were delivered in identical containers. The source of caffeine in e+ shot is green tea (*Camellia sinensis*) and yerba mate (*Ilex paraguariensis*) and because e+ shot contains all-natural ingredients there are some variations in content of caffeine; in this study the batch of e+ shot contained approximately 85 mg caffeine. The proprietary blend of adaptogenic herbs present in e+ shot is listed in Table 1, with Table 2 providing a comparison of the bioactive components of the three beverages. The placebo and active comparator were prepared in the same base components as e+ shot (purified water, apple juice concentrate, glycerin, pomegranate juice concentrate, natural flavors, malic acid, potassium sorbate, and sodium citrate) to which either 0 mg (placebo) or approximately 98 mg synthetic caffeine (active comparator) were added. Quantitation of caffeine in the study products was verified according to Eurofins Scientific Inc. (Des Moines, IA, USA). There were no changes in methods after trial commencement.

### 2.2. Screening

Potential participants were recruited from (i) large university classes (>30 students), (ii) announcements on bulletin boards, and electronic listservs, (iii) flyers at local small businesses, and, (iv) through word of mouth, from 16 September 2015 to 9 August 2016. Participants were invited to complete screening questionnaires (medical history, diet, mood) administered online using SurveyMonkey Inc. (San Mateo, CA, USA, www.surveymonkey.com).

Participant exclusion criteria included: (i) were under the age of 18 or over the age of 45, (ii) a self-reported body mass index (BMI) > 30, (iii) above average feelings of energy (scores > 12) during the week prior to the screening using the vigor scale of the 30-item Profile of Mood States-Short Form (POMS-SF) questionnaire [29]. The rationale for this cut-off was that a “ceiling effect” may occur with this particular measure. Since the POMS-SF vigor scale only measures vigor scores up to 20, including subjects who already display above average feelings of energy would likely minimize any potential energizing influences of either caffeine or e+ shot. (iv) high caffeine consumers (>21 servings of 170.5–341 mL caffeine beverages per week) [30], (v) high consumption of polyphenols (>100 total combined servings of cocoa, caffeine, fruits or vegetables high in flavanols) [31], (vi) reported a chronic health condition requiring prescription or over-the counter medication (excluding contraception) on a continual basis, (vii) were pregnant or reported a chance of being pregnant, (viii) had a heart condition, high blood pressure, gastrointestinal (GI) tract disorder, bipolar disorder, or allergy to caffeine (x) were current smokers, and (xi) consumed nutritional supplements (i.e., herbs, vitamins, or creatine; not including supplementation of protein without caffeine).

### 2.3. Participants

Approval for the study was granted by the Clarkson University Institutional Review Board (approval # 16-34.1). The authors confirm that all ongoing and related trials for this intervention are registered on clinicaltrials.gov (#16-05). Volunteers not excluded by the screening were invited to the testing facility. All participants read and signed the approved consent form. Participants were informed that they would be taking part in a study investigating the effects of caffeine beverages on mental function, blood pressure, heart rate, and fine motor control.

Thirty (17 women and 13 men) participants completed the study. An a priori statistical power analysis was completed and showed that 24 participants would provide statistical power of 0.81 to detect a 2 Group × 4 Time interaction effect size of 0.65 given an α of 0.05 and assuming a correlation across the repeated measures on time of 0.70 [32]. To reduce potential for Type II errors, 30 participants completed the study in case there were outliers and data had to be excluded. No data were excluded from this study. Characteristics from the final sample (*n* = 30) are reported in Table 3.

The average reported nightly sleep, during the month prior to the study, was 7.6 ± 0.8 h. The number of hours of reported sleep the night before each of the three testing sessions did not significantly differ between conditions (*p* = 0.731); placebo (6.5 ± 1.3 h); caffeine (6.4 ± 1.1 h) and e+ shot (6.3 ± 1.2 h).

Participants appeared to be low consumers of caffeine (4.2 ± 3.8 servings per week) and polyphenols (126.74 ± 12.16 servings per month). All participants reported refraining from caffeine in the 24 h prior to testing.

### 2.4. Salivary Caffeine Collection

Salivary samples were obtained by passive drool using the SalivaBio collection system (Salimetrics, State College, PA, USA) and stored at −80 °C. Samples were collected at the start of each testing day and post-test session, to estimate the association between caffeine and changes in mood, blood pressure, heart rate, cognitive task, and fine motor performance. After all samples were collected, analysis was conducted at the Biochemistry and Proteomics Laboratory (Clarkson University, Potsdam, NY, USA). A secondary goal of salivary sampling was to determine to confirm the anticipated changes in caffeine status upon consumption.

### 2.5. Salivary Caffeine Analysis

Salivary caffeine concentrations were quantified using an established method and adapted from Nakano et al. [33] and Ptolemy et al. [34]. In brief, salivary samples (up to 600 μL) were thawed at room temperature and filtered through an Amicon concentrator (Millipore, Billerica, MA, USA) with a 3 kDa molecular weight cut-off filter followed by centrifugation at 10,000 rpm for 20 min. The retentate, containing proteins, and the flow-through containing caffeine were collected for analysis. Approximately 10 μL flow-through fraction was analyzed on a C-18 column using an Agilent Technologies 1260 Infinity HPLC system. Caffeine was quantitated using a 7-min run and a mobile phase consisting of 75% H_2_O/25% (*v*/*v*) acetonitrile with 0.05% (*w*/*v*) ammonium acetate and 0.2% (*v*/*v*) acetic acid at a flow rate of 0.75 mL/min. The caffeine peak was detected at 275 nm using synthetic caffeine standards prepared in both H_2_O and saliva with calibration determined using standards prepared from the same saliva sample. The caffeine peak had a retention time of 5.9 min.

### 2.6. Mental Performance Tests and Physiological Measures

The mental energy tests consisted of self-reported motivation and mood measures, and computerized cognitive tasks of sustained attention. These measures were chosen because they are consistent with a model of mental energy developed for nutrition researchers [35]. In addition, two serial subtraction tasks (Serial Threes and Serial Sevens) were included to facilitate comparisons with prior research on caffeine utilizing these protocols [36,37]. Consistent with prior caffeine research [37], we measured blood pressure and heart rate after the cognitive tasks and mood measures. Fine motor task performance was measured using the nine-hole peg test.

All cognitive testing was performed in a seated position in a thermoneutral (72 ± 0.8 °F), private lab setting, with sound attenuation and controlled lighting. Visual stimuli were presented that required a finger response. Participants used the keyboard to respond to information presented on 17^”^ screen on an Alienware laptop (17 R2 Model #P43F, Roundrock, Texas, USA). Prior to each cognitive task, the participants were given on-screen instructions and asked to press the “enter” key if they understood the directions or to get help from a researcher if they were uncertain. The tests were performed on the Membrain platform (PsychTechSolutions, Potsdam, NY, USA) using Java-coded software. The results from the tasks automatically downloaded into an Excel file and two research assistants independently manually re-arranged the data, for data analysis, and any discrepancies were resolved.

(1)*Serial Three and Serial Seven subtraction tasks*: Participants were asked to silently subtract backwards in three’s or seven’s from a random starting number between 800 and 999 that was presented on the computer screen. Participants were instructed to type their answer as quickly and as accurately as possible. The number was cleared after entry of the response and participants continued to subtract three or seven from their answer. The task was scored for the number of correct and incorrect responses and the total attempts. In the case of incorrect responses, subsequent responses were scored as correct if they were correct in relation to the new number. Participants were given an opportunity to complete as many attempts as possible in two minutes [36,37,38].(2)*Continuous Performance Task (CPT)*: Participants monitored a continuous series of letters (A–Z; Tahoma Regular font, size 20 pt) presented on the screen for 1000 ms. Participants were told to respond to the detection of the letter “X” only when it was preceded by the letter “A” by striking the left key on the key pad. The task was scored for percentage of target strings correctly detected, errors of omission (missed targets), average reaction time for correct detections, and the number of false alarms. The task lasted for two minutes and 48 targets were randomly presented [21,37].(3)*Rapid Visual Input Processing (RVIP) task*: Participants were required to monitor a continuous series of digits (1–9; Tahoma Regular font, size 20 pt). Each individual digit was presented for 1000 ms and the participant was given a primary, secondary, and tertiary task. The participant’s primary and secondary tasks were to detect the presentation of three successive odd and even digits that were all different (e.g., 9-3-7, 2-6-8), and the tertiary task involved the identification of a specific number (i.e., 6). The participants pressed the right key for primary and secondary responses and the left key for tertiary responses. The task was scored for the number of primary, secondary, and tertiary targets correctly detected, the average reaction time for correct detection of each target, the number of false alarms for each task, and errors of omission (missed targets). There were 16 primary target, 16 secondary targets and 96 tertiary targets. The task lasted 16 min and a total of 960 stimuli were presented during that time [36,38,39].(4)*Motivation to perform cognitive tasks:* Participants rated the intensity of their current motivation to perform mental tasks using a scale supported by validity evidence [21,38]. The 0–10 categorical scale ranges from “No motivation” (left end, scored as 0) to “Highest motivation imaginable” (right end, scored as a 10).(5)*Profile of Mood Survey-Short Form (POMS-SF)*: The 30-item POMS-SF was used to assess current mood states using a five-point scale ranging from “Not at all” (scored as 0) to “Extremely” (scored as 4). The tension/anxiety, depression, anger, fatigue and vigor are scored as a sum of five variables (i.e., tension = tense + shaky + uneasy + nervous + anxious) and can range from 0 to 20. Confusion has a variable subtracted from it (i.e., confusion = confused + muddled + bewildered + forgetful − efficient) and can range from −4 to 16 [40]. Among healthy participants, the Cronbach’s alpha, a measure of internal consistency, has been reported as 0.90 for the POMS-SF [41]. The Cronbach’s alpha for this current study was between 0.364 and 0.922 (vigor = 0.922, fatigue = 0.863, depression = 0.598, tension = 0.364, anger = 0.519, and confusion = 0.419).(6)*Mental and Physical State and Trait Energy and Fatigue Scales:* Participants rated their current feelings of mental and physical energy, and fatigue, using a three-part scale supported by validity evidence [21,38,42,43]. The states scale is a 12-item measure of the intensity of current mental energy, mental fatigue, physical energy, and physical fatigue moods. For each state, responses to three items were summed to provide a measure of mental and physical energy or fatigue [44]. The scales require the use of a 0–100-point visual analog scale (VAS) however, due to limitations in data collection techniques, the scale was modified to a 0–10 Likert scale anchored by “absence of feelings” (left end, scored as 0) and the “strongest intensity of feelings” (right end, scored as 10). This modification is the same as Boolani et al. previously cited [21,43]. Among healthy adults the Cronbach’s Alpha was reported to be between 0.89 and 0.91 [44]. The Cronbach’s alpha for this current study was between 0.707 and 0.874 (state physical energy = 0.785, state physical fatigue = 0.837, state mental energy = 0.707, and state mental fatigue = 0.874).(7)*Blood Pressure:* Blood pressure was measured using a digital blood pressure monitor (Omron 3 Series; model: BP710N, Omron Health Care Inc, Forest, IL, USA) on the participant’s right upper-arm [45,46].(8)*Heart Rate:* Heart rate was measured using a pulse oximeter (Veridian Deluxe; model 11-50D, Veridian Healthcare, Gurnee, IL, USA) on the participant’s right index finger.(9)*Nine-hole peg test*: The validated nine-hole peg test of finger dexterity was used to measure fine motor control [47,48]. The 12 cm × 12 cm wooden pegboard contained nine holes and was placed on the desktop in front of the seated participant. There were nine 0.64 cm wide cylindrical pegs, were placed on the desktop outside of the container on the right side of the board and on the left side of the board for when the participant’s right hand and left hand were tested, respectively. Participants were instructed to place one peg at a time into the pegboard holes until they were filled, and then remove each peg one at a time onto the desktop as fast as they could, first with their dominant hand and next with their non-dominant hand. Each test was performed twice.

### 2.7. Procedure

The study consisted of a familiarization day (~1 h) followed by three testing days (~3 h). The familiarization day start time was between 6–8 a.m. and was chosen and agreed upon by each participant. Participants were instructed to refrain from caffeine and alcohol consumption starting the night before the three testing days and advised to get their typical amount of sleep.

Familiarization Day: To reduce experimental error that may occur due to learning effect, participants were asked to come to the lab for a practice session where they completed a single trial run of all the daily assessments. This data was not included in any analyses. For participants’ characteristics, height was measured using a stadiometer and weight was measured using a digital scale (Tanita TBF-410, Tanita Corporation, Tokyo, Japan).

Testing days 1–3: Using randomizer.org participants were randomly assigned to the order in which the beverage would be administered before the familiarization day. Participants were tested between 6–8 a.m. and each testing day started within ±30 min of their familiarization day to account for potential diurnal variations [49]. Since sleep loss has substantial effects on mood and cognition [49], participants who reported two hours more or less than their usual sleep duration (reported during the screening) were not tested that day and rescheduled, as were those who reported drug use or the consumption of caffeine containing beverages or foods the night before testing. Each testing day was scheduled a minimum of 48 h after the previous one. Upon reporting to lab, participants completed an initial screening to ensure that they followed the pre-testing instructions. Participants completed a series of surveys that asked them about their previous night’s sleep, food, beverage, and drug consumption over the prior 24 h.

After participants were screened for eligibility on the day of testing, they were asked to accumulate saliva at the bottom of their mouth and use a plastic drinking straw to drool 2 mL of saliva into a 10 mL test tube. Baseline measures of sustained attention (the mental energy test battery), motivation, mood, blood pressure (BP), heart rate (HR), and fine motor task performance were obtained next (Figure 1). After the completion of baseline testing, participants were administered the beverage and instructed to consume it within two minutes. Following the administration of the beverage, participants were given a 28-min break but were not allowed to participate in strenuous physical or mental activity or consume additional snacks or beverages. Three additional 27-min mental energy battery tests were completed and punctuated by 10-min rest breaks as shown in Figure 2. Finally, participants provided a post-test saliva sample using the drool down method. The timing of the mental energy test battery, mood and physiologic measures is presented in Figure 3.

### 2.8. Data Treatment and Statistics

#### 2.8.1. Preliminary Analyses

Questionnaire data from SurveyMonkey and cognitive data were downloaded into Microsoft Excel. All data were exported into IBM SPSS (Version 25.0) for analysis. All statistical analyses were performed prior to breaking the blind. Scatterplot, descriptive statistics, skewness, kurtosis, and normality (Wilks–Shapiro tests *p* < 0.05) were assessed. Values deemed erroneous, most likely due to coding errors for some of the cognitive tasks were removed. The post-treatment minus pre-treatment changes in salivary caffeine among the three treatments were examined using t-tests to determine whether treatments influenced salivary caffeine concentrations in expected ways (i.e., salivary caffeine increasing after caffeine consumption). To test for desired mood outcomes after the performance of the cognitive task intervention, we analyzed mood data immediately prior to and after the first set of cognitive tasks on all three days. Statistically significant differences were noted between anger, energy, fatigue, and mental energy (*p* < 0.05) thus validating that the 22-min cognitive task batter was sufficient for causing changes in mood states. No participants had pre-testing salivary caffeine levels > 0.05 µg/mL suggesting that all participants followed instructions to abstain from caffeine prior to testing day. Additionally, we measured cortisol levels pre and 136 min post-consumption. We used an a priori test-retest reliability of 0.90. Since our cortisol analyses did not meet our a priori test-retest reliability measure we did not include them in our primary analysis.

#### 2.8.2. Primary Analyses

Significant interactions between beverages were tested using Repeated Measures Analysis of Covariance (ANCOVA) while controlling for prior night’s sleep [21,38,50]. Only one significant outlier was detected, the inclusion of which was shown to have no effect on significance. Variables that exhibit a poor fit were not removed or transformed. The justification for this comes from the nature of the repeated measures format where transformations to one measure must be evenly applied to all other measures of that variable which would almost exacerbate analytical issues and make interpretation more difficult. Ordinal variables such as the POMS-SF scores were approximated as continuous but some ordinal variables and some ration variables had very few occurring values in the data making any ANOVA-type testing unreliable. These variables often displayed large violations of homoscedasticity or were poorly approximated as continuous and results from these variables have been marked. Despite this they have not been discarded because of the lack of a nonparametric alternative to Repeated Measures ANCOVA in addition to previously mentioned reasons. A two-treatment by four time point ANCOVA were performed on all combinations of all collected variables. Violations of sphericity were accounted for using the Hyunh–Feldt correction to degrees of freedom. Additionally, two-treatment by two time point ANCOVAs were also performed on all combinations of all collected variables, testing both between the pre-treatment and first post-treatment time points and between the pre-treatment and last post-treatment time points. Tukey’s analysis of marginal means was used to determine which measures were different for variables that exhibited significant. The final research question sought to determine whether e+ shot differed from placebo and a caffeinated dose-matched active comparator on mood, blood pressure, cognitive and fine motor task performance. Since our a priori power analysis and eventual subject size was powered for a two-group by four time point comparison and not powered for a three-group by four time point comparison, we used multiple repeated measures ANCOVAs. Following the suggestions of several authors [51,52,53,54], this study presents both unadjusted and Bonferroni adjusted *p*-values for hypothesis tests. Unadjusted significance tests are included considering this study’s goal to compare e+ shot to just placebo and caffeine and to usew the caffeine and placebo comparison only as a control to confirm that subjects had the desired effects to caffeine. Concerns that adjustments for multiple comparisons can lead to erroneously heightened Type II error rates [51,52,53,54] also lead us to report both unadjusted and adjusted significance. To present a more conservative estimate that highlights the possibility of false positives associated with repeated comparisons, a Bonferroni correction was applied, and unadjusted *p*-values multiplied by 3 (adjusted for the 3 different comparisons) were used to determine Beverage × Time interaction effects in the 2 × 4 ANCOVA. These results are reported in the manuscript and in either case, readers should be mindful of effect sizes of the results alongside the results of either hypothesis tests. An α of 0.05 was used.

## 3. Results

After screening 1035 surveys, 30 participants completed the study (see Figure 4). Recruitment and data collection lasted from 25 September 2015 to 10 December 2016 until 30 participants had completed all three days of treatment. There were no harms or unintended consequences for any of the interventions.

### 3.1. Salivary Analysis

As expected salivary caffeine levels increased significantly upon consumption of either caffeine (mean change = 5.21 µmol L^−1^; t = −9.652, df = 26, *p* < 0.001) or e+ shot (mean change = 4.46 µmol L^−1^; t = −8.557, df = 27, *p* < 0.001), while no significant difference was noted between conditions (*p* > 0.05). Means and standard deviations for all outcomes are presented in Table 4.

### 3.2. Placebo vs. Active Comparator

Feelings of vigor were found to significantly increase (F(1, 57) = 4.028, η^2^ = 0.066, unadjusted *p* = 0.050, adjusted *p* = 0.15) in the active comparator (caffeine) ($\bar{\Delta }=1.10$), while the placebo had a slight decline ($\bar{\Delta }=-0.04$) 30 min post-consumption. Caffeine also significantly decreased feelings of confusion (F(1, 57) = 6.670, η^2^ = 0.105, unadjusted *p* = 0.012, adjusted *p* = 0.036) ($\bar{\Delta }=-0.53$)and the total mood disturbance (F(1, 57) = 5.136, η^2^ = 0.085, unadjusted *p* = 0.012, *p* = 0.036) ($\bar{\Delta }=-1.50$) compared to placebo ($\bar{\Delta }=0.07$,$\bar{\Delta }=0.27$), at 30 min post-consumption. Caffeine was associated with a significant increase in both diastolic (F(1, 57) = 11.525, η^2^ = 0.168, unadjusted *p* = 0.001, adjusted *p* = 0.003, caffeine $\bar{\Delta }=5.10$, placebo $\bar{\Delta }=0.30$) and systolic blood pressure at 30 min (F(1, 57) = 10.861, η^2^ = 0.160, unadjusted *p* = 0.002, adjusted *p* = 0.006, caffeine $\bar{\Delta }=5.30$, placebo $\bar{\Delta }=-2.27$) post consumption.

Analysis also yielded significant improvements in the RVIP secondary task percent correct (F(1, 36) = 4.98, η^2^ = 0.122, unadjusted *p* = 0.032, adjusted *p* = 0.096) and percent omission (F(1, 36) = 4.789, η^2^ = 0.117, unadjusted *p* = 0.035, adjusted *p* = 0.105) for caffeine ($\bar{\Delta }=3.65$, $\bar{\Delta }=-2.57$) when compared to the placebo ($\bar{\Delta }=-4.00$, $\bar{\Delta }=4.57$), 30 min post consumption. Placebo consumption was associated with a significant decline ($\bar{\Delta }=-4.83$) in CPT percent correct F(1, 51) = 4.225, η^2^ = 0.077, unadjusted *p* = 0.045, adjusted *p* = 0.135) and a corresponding increase ($\bar{\Delta }=4.99$) in CPT percent omission (F(1, 51) = 4.236, η^2^ = 0.077, unadjusted *p* = 0.045, adjusted *p* = 0.135), while caffeine consumption exhibited no change in either ($\bar{\Delta }=0.63$, $\bar{\Delta }=-0.53$), 90 min post consumption.

The 2 × 4 ANCOVA yielded significant differences in CPT percent incorrect; however, those results were rejected due to poor homoscedasticity of the residuals. A significant difference (F(3, 78) = 3.561, η^2^ = 0.120, unadjusted *p* = 0.018, adjusted *p* = 0.054) was noted for the RVIP tertiary percent omission where the caffeine promoted a large decrease 30 min post consumption followed by tapering off. For systolic blood pressure, when participants consumed caffeine they experienced a significant (F(3,171) = 4.216, η^2^ = 0.069, unadjusted *p* = 0.007, adjusted *p* = 0.021) spike 30 min post consumption with tapering off after, compared to the placebo, which noted an initial drop in systolic blood pressure before values increased to pre-consumption levels.

#### 3.2.1. e+ Shot vs. Placebo

A significant difference (F(1, 56) = 6.066, η^2^ = 0.098, unadjusted *p* = 0.017, adjusted *p* = 0.051) in systolic blood pressure was observed when comparing the e+ shot ($\bar{\Delta }=2.60$) and placebo ($\bar{\Delta }=-2.27$) between pre-consumption and 30 min post-consumption. A significantly greater increase (F(1, 56) = 5.197, η^2^ = 0.085, unadjusted *p* = 0.026, adjusted *p* = 0.078) in serial subtract 3 attempts was noted for placebo ($\bar{\Delta }=6.54$) compared to e+ shot ($\bar{\Delta }=2.43$) between pre-consumption and 30 min post-consumption.

#### 3.2.2. e+ Shot vs. Active Comparator

When compared to e+ shot, the comparator elicited a greater ($\bar{\Delta }=3.83$) increase in serial subtract 3 attempts compared to the e+ shot ($\bar{\Delta }=2.43$) (F = (1, 55) = 10.123, η^2^ = 0.156, unadjusted *p* = 0.002, adjusted *p* = 0.006), 30 min post-consumption. Serial subtract 3 total attempts also exhibited significance (F(3,162) = 2.867, η^2^ = 0.050, unadjusted *p* = 0.038, adjusted *p* = 0.114) when considering all four time points, with e+ shot showing steady upward trends in total attempts while caffeine promoted a significant increase 30 min post-consumption with subsequent tapering off (Figure 5). Significant differences (F(1, 57) = 4.917, η^2^ = 0.081, unadjusted *p* = 0.031, adjusted *p* = 0.093) were also noted in diastolic blood pressure when comparing e+ shot to caffeine 30 min post-consumption.

A sharp increase ($\bar{\Delta }=5.10$) in diastolic blood pressure was observed in participants consuming caffeine while a lower increase ($\bar{\Delta }=0.80$) was noted in participants who consumed the e+ shot.

Comparing pre-consumption and 90 min post-consumption, participants who consumed the e+ shot saw a significant (F(1, 45) = 6.162, η^2^ = 0.128, unadjusted *p* = 0.014, adjusted *p* = 0.042) decrease $\bar{\Delta }=49.89$) in CPT reaction time compared to caffeine which noted a small change in the opposite direction ($\bar{\Delta }=-25.42$).

#### 3.2.3. Correlation between Changes in Caffeine and Changes in Mood and Cognition

Changes in salivary caffeine was significantly associated with changes in serial subtraction three attempts (r = 0.382, *p* = 0.045). All other relationships were weak or were not significant.

## 4. Discussion

To our knowledge, this randomized, double-blinded, placebo and comparator-controlled study design is among the first to systematically compare the acute effects of caffeine alone vs. a caffeine-containing, adaptogenic-rich energy shot on a battery of mental performance and cognitive parameters. Our findings suggest that several of the stimulatory effects of caffeine may be modulated by the adaptogenic herbs present in the energy shot (e+ shot), whereas other observations suggested the adaptogenic herbs may directly antagonize some of caffeine’s (neuro)physiological effects. Collectively, these results support the long-standing characterization of adaptogenic herbs as modulators of mental stress as described by several investigators [25,28].

### 4.1. Active Comparator vs. Placebo

Caffeine alone resulted in improvements in feelings of vigor approximately 30 min post consumption when compared to the placebo, which is in line with prior research [55], but subsequent improvements were smaller and not significant. Mean confusion scores did not significantly change with the caffeine intervention, which is also consistent with prior studies [21,56]; however, we also noted a significant interaction because confusion increased with the placebo, a finding also consistent with prior work [21]. We speculate that the confusion scores increased in response to the stress of performing 108 min of sustained cognitive tasks (4 × 27 min) across a 3-h testing session. We also noted minimal improvements in caffeine on cognitive task performance for both the CPT and RVIP, which is what we expected based on prior research [55]; however, when we examined the interaction between both treatments the decline in cognitive task performance with the placebo yielded statistically significant results. This suggests that caffeine could reduce the impact of mental task performance on high event (CPT) and low event (RVIP) tasks, consistent with other reports [6,57]. Caffeine alone also resulted in increases in systolic and diastolic blood pressure similarly shown by others [5,58,59,60,61]; nevertheless, this increase in blood pressure did not result in any hypertension in our healthy participants. There were negligible declines in both systolic and diastolic blood pressure with the placebo condition. These results confirm that our subjects had the expected response to caffeine.

#### 4.1.1. e+ Shot vs. Placebo

The interaction between the e+ shot energy drink and placebo resulted in a significant difference in systolic blood pressure 30 min post-consumption. This may be explained by the fact that e+ shot promoted a slight increase in systolic blood pressure while the placebo was associated with a decline. Similar to that noted above with caffeine, the change in systolic blood pressure in response to e+ shot consumption was still within normotensive range. There was no significant difference after 30 min with blood pressure as the placebo condition returned to pre-testing levels, and there were negligible increases in blood pressure with e+ shot. An interesting finding was that 30 min post consumption there was a significantly greater increase in serial subtraction 3 attempts in the placebo group compared to e+ shot. However, this may be explained by the fact that the number of attempts prior to the consumption of placebo was much lower than on the day when participants consumed the e+ shot energy drink. Alternatively, this finding may have simply been the result of regression to the mean.

#### 4.1.2. e+ Shot vs. Active Comparator

The caffeine beverage noted improvements 30 min post consumption on most cognitive measures, vigor and mental energy, with these improvements tapering out at 69 and 108 min. The e+ shot energy drink exhibited slower improvements in cognitive measures, vigor, and mental energy with both beverages noting similar improvements in cognitive measures, vigor, and mental energy (as noted by increased attempts on the serial subtract 3 task) from baseline to 108 min post beverage consumption. There was a significant difference between serial subtraction 3 attempts between the two beverages after the first 30 min with the number of attempts tapering off with the comparator while the measures continued to increase 68- and 108-min post consumption for e+ shot (Figure 5). The same is true for increases in systolic and diastolic blood pressure. These results suggest that the caffeine in both e+ shot and the comparator beverage is having its anticipated effects. In contrast, however, the adaptogens in e+ shot may have modulated these responses observed with caffeine only. These results are in line with prior studies that have examined the effect of adaptogens on human performance [26,28,62,63,64]. If the study had been conducted for a longer period of time (i.e., 2.5–3 h post consumption) and the trend in improvements in cognitive task performance, vigor, and mental energy continued we may have noted a similar trend in enhanced alertness and mental energy as that recently reported by Srivastava and colleagues who also compared several treatments including a caffeinated comparator, an herbal preparation, and the herbal preparation containing caffeine and noted similar multifactorial interactions on measures of mental alertness and sustained attention [65].

### 4.2. Possible Mechanisms

Mechanistically, adaptogens (or their primary components) have been reported to modulate stress-induced cellular pathways involving the stress hormones cortisol [66], eicosanoids [67], cerebral energy metabolism [68], and neurotransmitters [69,70,71], to name a few molecular targets. Potentially, any biological or metabolic interactions between caffeine and adaptogens could promote either antagonistic, additive, or synergistic cellular or physiologic responses. Subsequently, these interactions may have contributed to the differential responses observed from our results, particularly as all our mental tasks implicitly involve neurotransmission and energy homeostasis. For example, caffeine is known to modify neurotransmission by effecting dopamine synthesis, release of noradrenaline, as well as by enhancing serotonin [72,73,74]. Similarly, some adaptogenic herbs have also been reported to enhance status of each of these neurotransmitters in the brain [68,69,75]. In contrast, while caffeine inhibits phosphodiesterase, resulting in enhanced cyclic AMP concentrations, adaptogens have often been observed to reduce cyclic AMP concentrations via down-regulation of adenylate cyclase, and the up-regulation of phosphodiesterase [68]. Moreover, under many conditions, caffeine also stimulates glutamate and acetylcholine biosynthesis primarily via adenosine A1 antagonism [76,77], whereas adaptogens are associated with a reduction in glutamate excitotoxicity [78].

Another mechanism by which herbs could influence caffeine-dependent effects is via genetic or cellular regulation of caffeine metabolism. Caffeine is primarily metabolized via CYP1A2 an enzyme with significant interindividual variability attributed to single nucleotide genetic polymorphisms (SNP) [79]. Many herbs, including some of the adaptogens in e+ shot have been demonstrated to influence this disposition. For example, Ho et al. [80] reported potent inhibition of CYP1A2 by crampbark (*Virburnum opolus*) using a microplate-based assay of cDNA expressed CYP isoforms. Similarly, *Schisandra chinensis* has also been found to variably modify CYP isoforms [81]. Finally, Sheriffdeen et al. [82] observed other traditional herbal medicines (not those found in e+ shot) inhibited CYP1A2-mediated metabolism of caffeine humans upon acute consumption of these herbs, most likely through competitive and/or non-competitive enzymatic inhibition. Collectively, these studies provide a plausible rationale for future studies to evaluate the pharmacokinetic effects of the adaptogenic herbs in e+ shot upon caffeine metabolism.

Clinically, we are unaware of any studies comparing the neurocognitive effects of caffeine vs. a caffeine, adaptogenic-rich energy shot. Nevertheless, there are a few studies that may provide some insight into the potential effects and interactions among adaptogens, caffeine, and/or other herbal components, in regard to mental performance and cognition. For example, Aslanyan et al. [83] conducted a double-blinded, placebo-controlled acute-dose study evaluating the mental performance effects of a well characterized adaptogenic product (ADAPT-232) containing a standardized dose of *Rhodiola rosea*, *Schizandra chinensis*, and *Eleutherococcus senticosus*. They reported that participants consuming ADAPT-232 exhibited improved attention, speed, and accuracy during cognitive testing. It should be noted, however, that other studies have not found a beneficial effect of adaptogens on mental performance [84]. Recently, a similar-type comparison to the current study was reported in which researchers compared the effect of *Alpinia galanga*, a traditionally consumed herb in Ayurvedic and Chinese medicine, alongside that of both a placebo and a caffeine-matched placebo in mental performance tasks [65]. The authors reported increased alertness in participants consuming the herb, whereas participants consuming the combination herb/caffeine product observed a significant reduction in mean response time. In that study, caffeine alone increased alertness followed by a decrease (“crash”) at 3 h. In contrast, the herb improved alertness up to 5 h. The combination had a compromised result, i.e., the combination product mitigated against the caffeine crash. These researchers hypothesized that the herb enhanced dopaminergic activity/availability similarly to that demonstrated by caffeine. This hypothesis is consistent with an increase in dopamine brain levels in response to adaptogen exposure as noted above. The authors overall conclusion was generally consistent with our findings in noting that the combination of an herb/caffeine combination ameliorated any documented or perceived caffeine-induced crash.

### 4.3. Limitations

The current study had several features that may limit the generalizability of the findings. First, recruitment was limited to average or lower than average consumption of caffeine (<200 mg/day), fruits, vegetables, and other foods rich in polyphenol and POMS vigor scores of <13. Second, a relatively small number of participants were tested, and the timing and composition of the meals preceding testing were not controlled. Third, we did not obtain saliva samples between completion of beverage consumption and the second and third mental energy test battery, so it is unclear if caffeine and adaptogens were bioavailable prior to initiating the second mental energy test battery. We may have been able to determine the bioavailability of those substances throughout the study if we had measured saliva before each mental test battery. There is evidence that suggests that there was enough time between consumption of the shot and bioavailability of caffeine [85]; however, the bioavailability of caffeine when consumed with adaptogens has not been explored. It may be hypothesized that adaptogens limit caffeine’s bioavailability and reduced absorption rates of caffeine as noted by the significant spikes in mood 30 min after consumption of the caffeine only beverage, while gradual increases in mood were noted after the consumption of e+ shot. In addition, the caffeine dose was not administered relative to bodyweight, but was absolute (i.e., 85–100 mg caffeine), which limits direct comparison to studies that administer caffeine relative to body weight. Additionally, with the study design used, multiple statistical tests were conducted which increases the risk of one of the statistically significant results occurring by chance. To reduce this chance, we have reported both unadjusted and Bonferroni adjusted *p*-values. Finally, like other naturally caffeinated beverages (coffee, for example), e+ shot is a natural product with some natural variability in bioactive content and with potential stability heterogeneity even with strict cGMPs (Current good manufacturing practices). Per the testing specifications of e+ shot according to Isagenix International—the caffeine content contributed by green tea and yerba mate leaf extracts provides a range of 80–100 mg caffeine. The batch in question utilized for this clinical study contained approximately 85 mg. Simultaneously, we designed the caffeinated placebo to contain a similar content of caffeine within that range which, in this case, tested out as approximately 98 mg caffeine. While it would have been ideal to have been able to obtain the caffeine content for e+ shot prior to manufacture of the caffeinated placebo, this route was not available at the time of product manufacture. We do not believe, nor are we aware of any studies where this small difference in caffeine (85 vs. 98 mg) would contribute to any of the physiological or neurocognitive differential outcomes observed in the current study; particularly in the presence of other substantial, additional bioactives such as that present in e+ shot. Additionally, many of our non-significant findings may have been due to the violations of homoscedasticity of our data, which while trending towards significance did not show statistically significant results.

## 5. Conclusions

After controlling for variations in prior night’s sleep duration, the adaptogenic-rich energy beverage was noted to modulate the acute influence of caffeine on mental performance and cognitive parameters, as well as on mood and blood pressure. While the caffeine only condition noted increases in most aspects of cognition, mood and blood pressure 30 min after consumption, there was a tapering off effect at 68 and 108 min. Conversely, the adaptogenic rich blend also noted improvements over the 108-min span; however, these improvements were more gradual. The mechanisms by which these effects were modulated has yet to be elucidated and bioavailability of caffeine when consumed with adaptogens may play a role in these gradual improvements. Future research should compare the effects for an extended period of time (2.5–3 h post-consumption), to determine whether the trends persist.

## 6. Declarations

### Ethics Approval and Consent to Participate

Approval for the study was granted by the Clarkson University Institutional Review Board (approval # 16-34.1). All participants read and signed an informed consent form.

Availability of data and materials: The dataset supporting the conclusions of this article is available in the Mendley repository. The DOI number for this dataset is 10.17632/3s8sr9zth9.1

Competing Interests: Author EG is currently an employee of Isagenix International, LLC. Authors AB and CD received funding to conduct this study.

## Figures and Tables

**Figure 1 nutrients-12-01922-f001:**
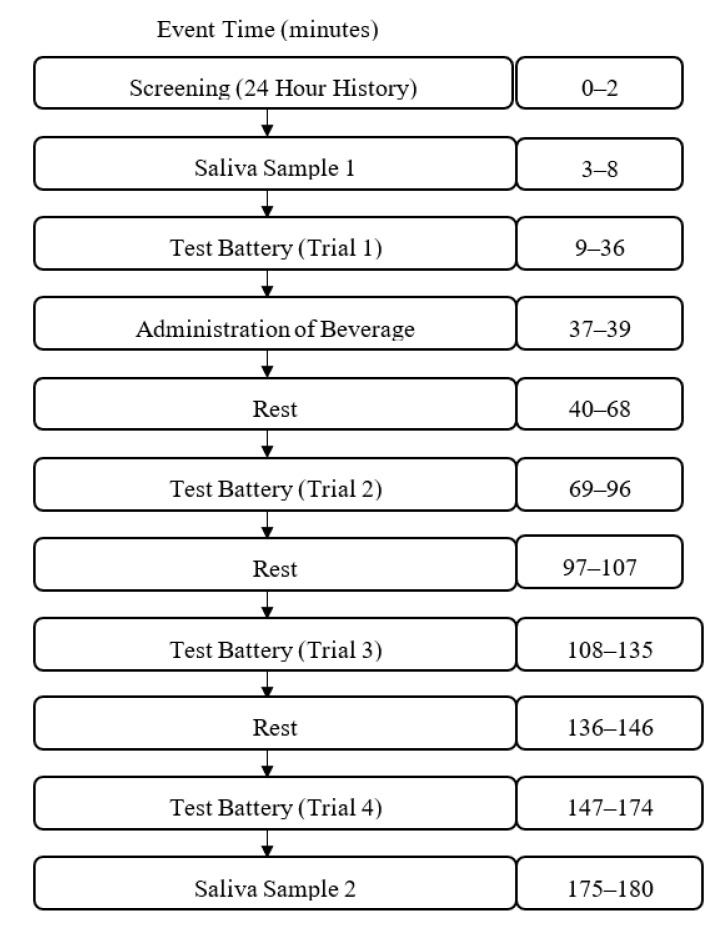
Test day process flow diagram.

**Figure 2 nutrients-12-01922-f002:**
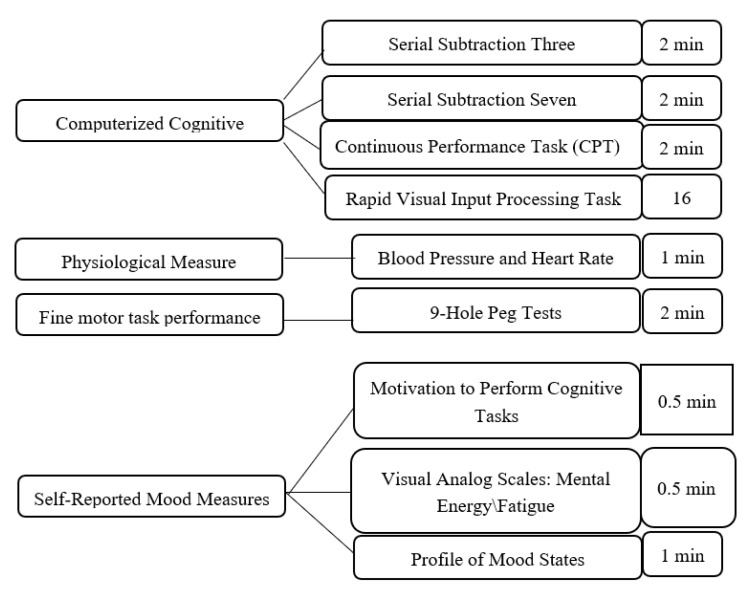
Tests and measures.

**Figure 3 nutrients-12-01922-f003:**
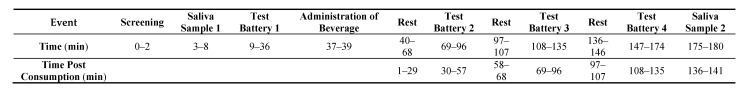
Test day timeline.

**Figure 4 nutrients-12-01922-f004:**
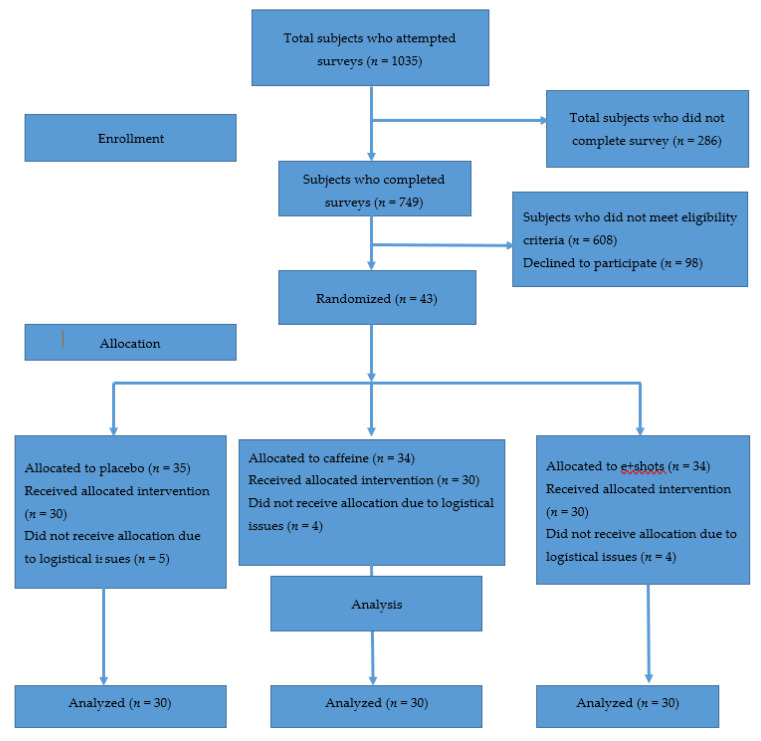
CONSORT flow diagram.

**Figure 5 nutrients-12-01922-f005:**
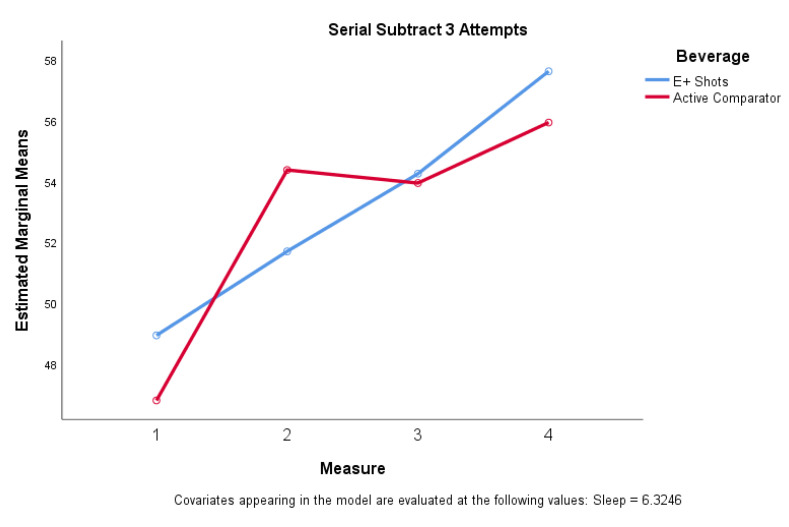
Serial subtraction 3 attempts for e+ shot vs. active comparator (caffeine).

**Table 1 nutrients-12-01922-t001:** Adaptogenic herbs contained in e+ shot (mg).

Ingredient	Quantity(mg)
Eleuthero (*Eleutherococcus senticosus*)	79
Hawthorn (*Crataegus oxycantha*)	59
Mountain ash (*Sorbus aucuparia*)	59
Cramp bark (*Viburnum opolus*)	59
Leuzea (*Rhaponticum carthamoides*)	40
Rhodiola (*Rhodiola rosea*)	20
Japanese aralia (*Aralia mandchurica*)	20
Licorice (*Glycyrrhiza yuralensis*)	20
Schizandra (*Schisandra chinensis*)	20
Chaga mushroom (*Inonotus obliquus*)	20

**Table 2 nutrients-12-01922-t002:** Comparison of Study Product Bioactives.

Treatment	Caffeine (Mg)	Adaptogenic Herbal Blend (Mg)
Placebo	0	0
Active Comparator (Caffeine)	98 (synthetic)	0
e+ shot	85.4 (green tea-*Camellia sinensis* and yerba mate-*Ilex paraguariensis* leaf extract)	2127

**Table 3 nutrients-12-01922-t003:** Participant characteristics.

Sex (Males/Females)	13/17
Age (years)	21.8 ± 4.4
Height (cm)	169.6 ± 12.4
Weight (kg)	67.6 ± 11.0
Body Mass Index (kg/m^2^)	23.5 ± 2.5
Race	
White	21
Asian	4
Black	4
More than one race	1
Amount of sleep on a typical night in the past month (h)	7.6 ± 0.8
Consumption of high-flavanol foods or beverages during the past month	
Caffeine drinks (servings)	4.2 ± 3.8
Cocoa (servings)	0.7 ± 1.3
Fruits (servings)	12.3 ± 12.4
Vegetables (servings)	25.1 ± 14.5

**Table 4 nutrients-12-01922-t004:** Mean (SD) for all measures.

Beverage	e+ shot				Placebo				Active Comparator			
Measure	Pre	Post 1	Post 2	Post 3	Pre	Post 1	Post 2	Post 3	Pre	Post 1	Post 2	Post 3
**Self-reported measures**												
Task motivation	6.03 (2.31)	6.33 (1.94)	5.97 (2.07)	6.42 (2.42)	5.93 (2.26)	6.17 (2.15)	5.90 (2.43)	5.97 (2.20)	5.87 (2.30)	6.67 (2.06)	6.13 (2.21)	6.11 (2.57)
POMS Vigor	8.80 (4.60)	9.67 (4.03)	9.76 (4.27)	10.15 (4.54)	9.07 (4.63)	9.03 (4.17)	9.33 (4.33)	9.38 (4.59)	8.93 (4.60)	10.13 (3.91)	9.87 (4.38)	9.89 (5.22)
POMS Fatigue	8.00 (2.89)	7.07 (2.48)	7.66 (3.27)	7.77 (3.15)	8.03 (2.94)	8.13 (2.98)	8.23 (2.62)	7.72 (2.55)	8.17 (2.81)	7.67 (2.94)	7.57 (2.81)	8.07 (3.22)
POMS Depression	5.50 (1.41)	5.50 (1.20)	5.45 (1.27)	5.31 (1.01)	5.37 (1.07)	5.43 (1.04)	5.30 (0.84)	5.28 (0.65)	5.30 (0.65)	5.23 (0.63)	5.20 (0.61)	5.25 (0.59)
POMS Tension	5.63 (1.16)	5.73 (1.34)	5.66 (1.52)	5.62 (1.10)	5.43 (0.73)	5.40 (0.77)	5.40 (0.72)	5.52 (0.91)	5.60 (1.30)	5.77 (1.28)	5.87 (1.25)	5.64 (1.03)
POMS Confusion	2.73 (1.57)	2.33 (1.45)	2.59 (1.57)	2.58 (1.60)	2.50 (1.25)	2.57 (1.17)	2.50 (1.04)	2.34 (1.05)	2.70 (1.54)	2.17 (1.21)	2.43 (1.43)	2.29 (1.56)
POMS Anger	5.47 (1.38)	5.33 (0.71)	5.41 (1.35)	5.27 (0.53)	5.27 (0.79)	5.30 (0.65)	5.60 (1.04)	5.31 (0.85)	5.90 (1.27)	5.53 (1.31)	5.47 (0.94)	5.57 (1.20)
Total Mood Disturbance	18.53 (9.39)	16.30 (8.42)	17.00 (9.15)	16.38 (8.83)	17.53 (8.75)	17.80 (7.67)	17.70 (7.60)	16.79 (8.16)	18.73 (8.39)	16.23 (7.52)	16.67 (8.08)	16.93 (9.28)
State Physical Energy	9.50 (6.55)	7.93 (6.11)	8.62 (6.38)	8.27 (5.84)	9.13 (5.87)	8.43 (5.55)	8.63 (5.37)	8.31 (5.55)	8.67 (6.14)	8.67 (6.23)	8.83 (6.05)	8.84 (6.58
State Physical Fatigue	14.63 (6.45)	15.97 (6.83)	15.31 (6.56)	15.31 (6.63)	13.87 (6.34)	14.23 (6.35)	14.57 (6.36)	14.69 (6.89)	16.40 (6.89)	16.40 (7.38)	16.38 (7.27)	16.92 (7.24)
State Mental Energy	10.27 (6.26)	8.70 (6.51)	9.17 (6.66)	9.73 (6.75)	9.93 (6.01)	9.23 (6.30)	9.90 (6.38)	8.83 (5.42)	10.13 (7.39)	9.43 (7.23)	9.52 (7.16)	10.16 (7.71)
State Mental Fatigue	13.40 (6.75)	14.60 (7.12)	14.07 (7.10)	14.46 (7.12)	13.27 (6.59)	13.30 (6.20)	13.67 (6.77)	13.90 (7.39)	15.13 (6.58)	15.60 (7.31)	15.34 (7.49)	16.44 (8.30)
**Physiologic measures**												
Systolic Blood Pressure	111.00 (16.18)	113.60 (13.74)	114.57 (16.45)	114.67 (14.21)	111.80 (12.22)	109.53 (12.62)	111.30 (15.50)	112.30 (13.37)	109.63 (12.99)	114.93 (16.41)	113.40 (15.07)	112.80 (15.04)
Diastolic Blood Pressure	71.50 (9.44)	72.30 (7.91)	74.80 (17.78)	72.03 (10.32)	71.00 (8.51)	71.30 (8.12)	73.60 (12.14)	72.57 (8.02)	68.97 (9.04)	74.07 (7.70)	74.37 (8.20)	73.17 (8.45)
Heart Rate	70.13 (13.84)	65.23 (11.31)	67.90 (14.51)	64.47 (13.64)	72.50 (13.86)	70.10 (13.98)	68.00 (13.86)	66.83 (14.18)	69.80 (13.99)	67.37 (12.35)	67.10 (13.33)	63.83 (12.36)
Non-dominant Hand (Avg.)	19.27 (2.65)	18.88(2.83)	18.58 (2.55)	18.17 (2.93)	19.42 (2.94)	18.80 (2.45)	18.70 (2.76)	18.54 (2.94)	19.31 (3.40)	18.73 (3.23)	18.78 (3.38)	18.44 (3.37)
Dominant Hand (Avg.)	18.60 (2.45)	17.93 (2.09)	17.68 (2.07)	17.50 (2.20)	18.87 (2.64)	18.21 (2.50)	18.22 (2.86)	17.86 (2.75)	18.65 (2.93)	17.95 (2.79)	18.02 (2.95)	17.56 (2.84)
**Objective cognitive task measures**												
Serial Subtract 3 # Correct	46.27 (14.70)	48.70 (16.08)	51.37 (15.54)	54.59 (16.90)	44.80 (16.24)	51.17 (15.57)	52.63 (16.26)	53.38 (16.15)	44.23 (17.08)	52.97 (16.16)	51.82 (16.91)	52.17 (17.85)
Serial Subtraction 3 % Correct	95.69 (4.23)	95.76 (3.96)	95.57 (3.38)	93.96 (8.89)	96.35 (3.98)	96.98 (3.07)	96.21 (3.18)	95.09 (4.48)	94.43 (6.85)	96.22 (3.88)	92.78 (7.83)	93.79 (6.63)
Serial Subtract 3 # Attempted	48.20 (14.59)	50.63 (15.90)	53.50 (15.36)	57.21 (15.98)	46.23 (16.18)	52.77 (15.84)	54.47 (16.11)	56.10 (16.01)	46.27 (16.45)	54.86 (15.92)	54.34 (16.32)	55.33 (16.60)
Serial Subtract 7 # Correct	26.17 (10.30)	29.03 (10.81)	31.27 (11.59)	33.48 (11.96)	24.73 (10.70)	27.83 (10.70)	28.83 (11.81)	30.43 (11.43)	27.00 (11.23)	30.20 (10.95)	30.41 (11.84)	31.60 (13.15)
Serial Subtraction 7 % Correct	92.78 (7.99)	93.59 (8.08)	93.80 (6.60)	93.17 (6.07)	92.53 (8.80)	94.36 (4.98)	92.92 (8.42)	91.21 (8.74)	92.72 (11.29)	91.65 (8.13)	91.44 (7.21)	90.60 (9.18)
Serial Subtract 7 # Attempted	27.90 (10.13)	30.80 (11.25)	33.07 (11.38)	35.69 (12.02)	26.53 (10.19)	29.38 (10.71)	31.57 (12.09)	33.90 (12.11)	28.77 (11.04)	32.60 (10.99)	33.47 (11.60)	34.27 (13.02)
CPT Percent Correct	80.46 (29.50)	78.18 (27.49)	82.95 (27.88)	86.40 (16.47)	83.92 (25.53)	77.79 (25.49)	79.07 (23.83)	79.09 (24.73)	84.93 (17.69)	83.32 (21.01)	85.26 (22.34)	85.56 (14.23)
CPT % Incorrect	0.49 (0.79)	0.59 (0.80)	0.51 (1.13)	0.41 (0.58)	0.44 (0.51)	0.48 (0.48)	0.37 (0.82)	0.48 (0.74)	0.44 (0.58)	0.45 (0.71)	0.56 (0.78)	0.37 (0.63)
CPT % Omitted	19.54 (29.50)	21.81 (27.49)	17.05 (27.88)	13.60 (16.47)	16.08 (25.53)	22.21 (24.49)	20.21 (23.72)	20.91 (24.73)	14.97 (17.36)	16.68 (21.01)	14.73 (22.34)	14.44 (14.23)
CPT Reaction Time (ms)	1475.20 (76.56)	1527.81 (158.49)	1506.42 (91.91)	1525.09 (82.97)	1441.63 (304.19)	1515.54 (95.82)	1516.12 (111.05)	1498.28 (72.03)	1499.64 (102.770	1495.55 (85.770	1527.74 (122.18)	1474.22 (75.06)
RVIP Primary % Correct	60.78 (26.18)	67.16 (23.99)	70.27 (24.57)	64.65 (20.36)	59.45 (24.98)	61.79 (25.86)	65.18 (20.86)	62.52 (26.73)	57.18 (27.66)	64.70 (25.18)	67.13 (24.75)	71.38 (19.06)
RVIP Primary % Incorrect	9.40 (9.64)	6.58 (8.34)	6.47 (8.96)	7.27 (5.79)	8.70 (9.57)	9.55 (8.72)	7.33 (8.27)	6.75 (8.72)	8.65 (10.28)	6.50 (7.30)	9.88 (10.50)	7.54 (7.88)
RVIP Primary % Omitted	29.82 (20.67)	39.62 (25.39)	22.82 (20.58)	27.90 (19.70)	31.85 (22.01)	28.65 (24.00)	27.43 (19.29)	30.73 (26.00)	34.13 (24.30)	28.85 (25.07)	20.51 (16.77)	20.69 (17.00)
RVIP Primary Reaction Time (ms)	782.84 (99.82)	774.57 (71.210)	752.33 (85.85)	736.22 (49.30)	777.76 (93.09)	761.51 (74.29)	791.83 (55.05)	770.84 (71.41)	786.74 (121.13)	773.95 (75.76)	733.10 (76.58)	727.96 (79.60)
RVIP Secondary % Correct	55.24 (23.10)	59.98 (25.75)	57.05 (30.08)	61.56 (23.46)	58.75 (21.45)	54.75 (19.69)	54.32 (24.46)	61.11 (24.75)	55.28 (24.23)	58.93 (27.25)	58.56 (28.20)	63.98 (24.26)
RVIP Secondary % Incorrect	2.64 (5.37)	0.00 (0.00)	0.83 (2.56)	1.04 (2.85)	1.36 (3.19)	0.42 (1.86)	0.46 (1.96)	0.52 (2.08)	1.09 (3.81)	0.00 (0.00)	0.36 (1.71)	0.38 (1.78)
RVIP Secondary % Omitted	42.08 (22.55)	39.62 (25.39)	0.42 (29.88)	37.35 (22.83)	39.85 (20.24)	44.42 (19.59)	45.17 (24.55)	38.28 (24.45)	43.64 (24.28)	41.07 (27.25)	41.07 (28.36)	35.63 (24.38)
RVIP Secondary Reaction Time (ms)	646.66 (98.86)	601.89 (76.14)	634.83 (102.15)	633.80 (67.24)	610.31 (124.82)	627.79 (59.08)	628.47 (108.98)	649.46 (90.90)	612.76 (65.11)	604.52 (87.25)	621.75 (129.66)	598.90 (92.00)
RVIP Tertiary % Correct	88.89 (12.38)	88.21 (14.81)	87.77 (16.48)	88.73 (14.23)	83.76 (19.05)	84.05 (15.79)	85.57 (13.61)	88.09 (12.85)	86.45 (11.74)	90.61 (8.10)	85.90 (16.46)	89.59 (10.67)
RVIP Tertiary % Incorrect	3.10 (5.36)	1.81 (4.35)	1.62 (3.42)	1.94 (3.61)	4.68 (7.09)	2.57 (3.97)	3.05 (7.95)	2.92 (5.03)	2.40 (4.55)	2.98 (8.17)	3.07 (4.89)	1.52 (4.18)
RVIP Tertiary % Omitted	9.55 (11.72)	10.30 (14.14)	11.40 (16.71)	8.42 (13.00)	12.56 (16.49)	15.32 (15.85)	11.47 (11.41)	10.16 (13.55)	11.18 (10.17)	9.20 (11.32)	12.43 (15.63)	9.41 (10.60)
RVIP Tertiary Reaction Time (ms)	761.78 (60.50)	731.21 (44.10)	728.52 (46.69)	737.23 (49.38)	749.10 (56.08)	745.10 (61.01)	738.66 (56.00)	717.86 (40.95)	752.61 (61.42)	746.90 (53.46)	751.04 (55.61)	739.53 (58.21)

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
