# Peer review of "Caffeine-Containing, Adaptogenic-Rich Drink Modulates the Effects of Caffeine on Mental Performance and Cognitive Parameters: A Double-Blinded, Placebo-Controlled, Randomized Trial"

_nutrients, 2020, doi:10.3390/nu12071922_

Round 1
Reviewer 1 Report
- Line 57 and wherever applicable - please use the SI units.
- When providing the producer, please disclaim the country - even when it may be derived from the remaining data.
- I do not understand why the authors chose to supplement the active control group with different amount of caffeine than the exp group.
- What was the rationale behind excluding those who consume relatively high amounts of polyphenols?
- What exactly is the serving of caffeine or other products in the current study?
- "Tahoma Regular font, size 20" - 20pt?
- Please either present data always as .XXX or 0.XXX
- Were the three groups similar in terms of age, sex, BMI and other measures?
Author Response
TRANSMITTAL LETTER
Nutrients- 836371
“Caffeine-containing adaptogenic-rich drink modulates the effects of caffeine on mental performance and cognitive parameters”
Dear Dr. Miljanovic and Distinguished Reviewers,
Thank you for reviewing our manuscript. We appreciate the thorough reviews and valuable feedback. What follows is a point-by-point description of how we have addressed all the concerns expressed by the reviewers. Our responses are in red text and bullet-points below the original reviewers comments, which is in black text.
Line 57 and wherever applicable - please use the SI units.
Thank you for pointing out the inconsistencies in our reporting to us. We have corrected lines 59-60 to reflect mg/ml. We have also corrected line 87 to reflect ml instead of oz. Line 126 and 127 have also been changed from oz to ml. We were unable to find other places where we did not use SI units.
When providing the producer, please disclaim the country - even when it may be derived from the remaining data.
Thank you for pointing this out. On line 88, we have added USA as the country of the manufacturer of e+shots (Isagenix International, LLC). We have also added the country on line 110 for the Europfins Scientific Inc address.
I do not understand why the authors chose to supplement the active control group with different amount of caffeine than the exp group.
We appreciate the reviewers concern and have addressed this in our limitations section. The explanation is available on lines 611-623:
“Finally, like other naturally caffeinated beverages (coffee, for example), e+ shot is a natural product with some natural variability in bioactive content and with potential stability heterogeneity even with strict cGMPs (Current Good Manufacturing Practices). Per the testing specifications of e+ shot - according to Isagenix International - the caffeine content contributed by green tea and yerba mate leaf extracts provides a range of 80-100mg caffeine. The batch in question utilized for this clinical study contained approximately 85mg. Simultaneously, we designed the caffeinated placebo to contain a similar content of caffeine within that range which, in this case, tested out as approximately 98mg caffeine. While it would have been ideal to have been able to obtain the caffeine content for e+ shot prior to manufacture of the caffeinated placebo, this route was not available at the time of product manufacture. We do not believe, nor are we aware of any studies where this small difference in caffeine (85 vs 98mg) would contribute to any of the physiological or neurocognitive differential outcomes observed in the current study; particularly in the presence of other substantial, additional bioactives such as that present in e+ shot.”
What was the rationale behind excluding those who consume relatively high amounts of polyphenols?
We appreciate the reviewer’s concern. We excluded people who are high consumers of polyphenols because polyphenols have been known to be mood and cognitive enhancers. By eliminating people who were high polyphenol consumers over the past month we reduce the risk of including people with enhanced mood and/or cognitive function.
What exactly is the serving of caffeine or other products in the current study?
On line 87, we have reported the serving size for all the products used in the current study. Additionally, the caffeine content of the three interventions is provided in Table 2.
"Tahoma Regular font, size 20" - 20pt?
Thank you for pointing this out. We have corrected it to state 20pt on lines 211 and 221.
Please either present data always as .XXX or 0.XXX
Thank you for pointing this out. We are presenting all data as 0.XXX
Changes have been made on lines 241-243, 253-255, 358, 365, 367, 369, 416, 417, 474
Were the three groups similar in terms of age, sex, BMI and other measures?
Participant demographics are provided in Table 3. Also, for clarification, because we performed a cross-over design all subjects received each of the 3 treatments.

Reviewer 2 Report
In this manuscript, the authors reported that the adaptogenic-rich energy beverage can modulate the acute influence of caffeine on mental performance using a double-blinded, placebo controlled, randomized trial. The article is generally very well written, but the data provided to support the conclusions is rather limited.
- Mental performance and cognitive parameters are well known to be easily affected by surrounding environment. Here, “placebo” sample have important meaning. Therefore, the photographic images about placebo, e+shot and active comparator, will help the readers’ understanding.
- I could not find the Figure 1.
- I could not look the total of Figure 2’s gains.
- In this study, dietary habituation is important information. Therefore, the authors should add their data, at least recent one month, using e.g. self-administered diet history questionnaires.
- The information about effect of adaptogenic-rich energy beverage on the bioavailability of caffeine, such as the absorption rate, is needed for conclude this manuscript.
- In this manuscript, the authors cited 83 publications in the Reference section. Among them, publication at 2017 was most novel. From there onward, any relative aspects have not been reported?
Author Response
TRANSMITTAL LETTER
Nutrients- 836371
“Caffeine-containing adaptogenic-rich drink modulates the effects of caffeine on mental performance and cognitive parameters”
Dear Dr. Miljanovic and Distinguished Reviewers,
Thank you for reviewing our manuscript. We appreciate the thorough reviews and valuable feedback. What follows is a point-by-point description of how we have addressed all the concerns expressed by the reviewers. Our responses are in red text and bullet-points below the original reviewers comments, which is in black text.
In this manuscript, the authors reported that the adaptogenic-rich energy beverage can modulate the acute influence of caffeine on mental performance using a double-blinded, placebo controlled, randomized trial. The article is generally very well written, but the data provided to support the conclusions is rather limited.
- Mental performance and cognitive parameters are well known to be easily affected by surrounding environment. Here, “placebo” sample have important meaning. Therefore, the photographic images about placebo, e+shot and active comparator, will help the readers’ understanding.
We believe that the reviewer is asking for the presentation of the three interventions to the participants. On lines 100 and 101 we have added information as to how the treatment was delivered to the participants.
“Participants received the treatment in an unmarked white container with a black top. All treatments were delivered in identical containers.”
- I could not find the Figure 1.
Figure 1 is the CONSORT flow diagram. Although it was initially provided in our original manuscript submission we will ensure that it’s again attached with our edits.
- I could not look the total of Figure 2’s gains.
Figure 2 is the testing flow diagram. Perhaps the reviewer could clarify this comment as we are unsure what the reviewer is requesting regarding this Figure 2.
- In this study, dietary habituation is important information. Therefore, the authors should add their data, at least recent one month, using e.g. self-administered diet history questionnaires.
We agree that dietary habituation is important which is why we: 1) excluded high consumers of both caffeine and total polyphenol intake, and 2) estimated dietary intake of caffeine and polyphenol consumption for those accepted into the study. and we have reported their pre-study caffeine and polyphenol consumption in Table 3.
- The information about effect of adaptogenic-rich energy beverage on the bioavailability of caffeine, such as the absorption rate, is needed for conclude this manuscript.
Thank you for pointing this out. We have addressed this in our limitations section (lines 593-610) as well as under Possible Mechanisms (line 525-530). We also agree with the reviewer that we should include that in the conclusion of our manuscript. Therefore, we have added an additional statement in our conclusion (lines 640-641).
“and bioavailability of caffeine when consumed with adaptogens may play a role in these gradual improvements.”
- In this manuscript, the authors cited 83 publications in the Reference section. Among them, publication at 2017 was most novel. From there onward, any relative aspects have not been reported?
We have updated our references to include more recent publications
- Qi HY, Li L, Ma H. Cellular stress response mechanisms as therapeutic targets of ginsenosides. Med Res Rev. 2018;38(2):625‐ doi:10.1002/med.21450
- Panossian A, Seo EJ, Efferth T. Effects of anti-inflammatory and adaptogenic herbal extracts on gene expression of eicosanoids signaling pathways in isolated brain cells. Phytomedicine. 2019;60:152881. doi:10.1016/j.phymed.2019.152881.
- Wu R, Xiao Z, Zhang X, Liu F, Zhou W, Zhang Y. The Cytochrome P450-Mediated Metabolism Alternation of Four Effective Lignans From Schisandra chinensis in Carbon Tetrachloride-Intoxicated Rats and Patients With Advanced Hepatocellular Carcinoma. Front Pharmacol. 2018;9:229. Published 2018 Mar 14. doi:10.3389/fphar.2018.00229
- Sheriffdeen MM, Alehaideb ZI, Law FCP. Caffeine/Angelica dahurica and caffeine/Salvia miltiorrhiza metabolic inhibition in humans: In vitro and in vivo studies. Complement Ther Med. 2019;46:87‐ doi:10.1016/j.ctim.2019.07.024

Reviewer 3 Report
Overall, the paper is well-written and the idea is well-conceived. The study design is a strength for this study. Primary concerns center on the number of variables and tests, and whether these support the overall conclusions. To that point, it seems a lot is made of statistically insignificant results. The primary significant finding between e-shot and caffeine appears to be attempts on the serial subtract 3 attempts, including demonstrating this on Figure 5. However, the meaning of this finding is not clear - in other words, among all the variables tested, how does attempting more serial 3 subtractions, without regard to number correct, reflect a positive response within the context of all the other comparisons on direct cognitive assessment which are not statistically significant? The concern is that there is a finding among many insignificant findings, and that this is perhaps overly emphasized. I would like there to be more discussion of the overall lack of findings, particularly on those assessments that are tests (not questionnaires) and ought to be somewhat more objective measures of specific cognitive variables.
The other main issue is with Table 4 which presents the mean and SD for all the variables of interest over the 3 conditions (placebo, caffeine, e-shot). I applaud the authors for being inclusive of all their data. However, the table itself is overwhelming. Most of the numeric values are meaningless without more context. I would recommend considering options to re-organize the presentation of this information, including demonstrating significance of comparisons. This might be organized into separate tables for self-report, physiologic measures, and direct cognitive assessment. This would be easier for readers to review, and by showing significant comparisons within the table(s) there would be greater clarity and context.
In light of the lack of significant findings (especially for the direct cognitive assessments), would the authors posit that this was a measurement issue? Or some other explanation for the minimal significant findings? Could they suggest improvements for future research that might improve upon their study and increase the likelihood of meaningful significant findings?
Author Response
TRANSMITTAL LETTER
Nutrients- 836371
“Caffeine-containing adaptogenic-rich drink modulates the effects of caffeine on mental performance and cognitive parameters”
Dear Dr. Miljanovic and Distinguished Reviewers,
Thank you for reviewing our manuscript. We appreciate the thorough reviews and valuable feedback. What follows is a point-by-point description of how we have addressed all the concerns expressed by the reviewers. Our responses are in red text and bullet-points below the original reviewers comments, which is in black text.
Overall, the paper is well-written and the idea is well-conceived. The study design is a strength for this study.
Thank you for the kind words.
Primary concerns center on the number of variables and tests, and whether these support the overall conclusions.
We appreciate this concern. We would like to acknowledge that we tested a number of variables that are consistent with what has been proposed by O’Connor (2006) to measure mental energy and cognitive function for nutritional related claims. This methodology with this many variables and tests has also previously been used in multiple studies (see list below)
O’Connor PJ. Mental energy: developing a model for examining nutrition‐related claims. Nutrition reviews. 2006;64:S2–6.
Some studies that have used a similar protocol along with the variables and tests include:
- Haskell CF, Kennedy DO, Wesnes KA, Scholey AB. Cognitive and mood improvements of caffeine in habitual consumers and habitual non-consumers of caffeine. Psychopharmacology. 2005;179(4):813–25.
- Scholey AB, Kennedy DO. Cognitive and physiological effects of an “energy drink”: an evaluation of the whole drink and of glucose, caffeine and herbal flavouring fractions. Psychopharmacology. 2004;176(3–4):320–30.
- Maridakis V, Herring MP, O’Connor PJ. Sensitivity to change in cognitive performance and mood measures of energy and fatigue in response to differing doses of caffeine or breakfast. International Journal of Neuroscience. 2009;119(7):975–94.
- Scholey AB, French SJ, Morris PJ, Kennedy DO, Milne AL, Haskell CF. Consumption of cocoa flavanols results in acute improvements in mood and cognitive performance during sustained mental effort. Journal of Psychopharmacology. 2010;24(10):1505–14.
- Maridakis V, O’Connor PJ, Tomporowski PD. Sensitivity to change in cognitive performance and mood measures of energy and fatigue in response to morning caffeine alone or in combination with carbohydrate. International Journal of Neuroscience. 2009;119(8):1239–58.
- Boolani A, Lindheimer JB, Loy BD, Crozier S, O’Connor PJ. Acute effects of brewed cocoa consumption on attention, motivation to perform cognitive work and feelings of anxiety, energy and fatigue: a randomized, placebo-controlled crossover experiment. BMC Nutrition. 2017;3(1):8
To that point, it seems a lot is made of statistically insignificant results. The primary significant finding between e-shot and caffeine appears to be attempts on the serial subtract 3 attempts, including demonstrating this on Figure 5. However, the meaning of this finding is not clear - in other words, among all the variables tested, how does attempting more serial 3 subtractions, without regard to number correct, reflect a positive response within the context of all the other comparisons on direct cognitive assessment which are not statistically significant?
We understand that the differences in cognitive measures are neither intuitive nor obvious, so we appreciate the opportunity to clarify some of these concepts. We specifically measured serial subtraction 3 and 7 because these tasks measure task vigilance and mental energy. Task vigilance in this context is defined as the number of correct responses and/or the percent correct response (both of which have been used in pervious literature (Scholey, 2010, Boolani, 2017). As a distinction, mental energy here is defined as the number of attempts when performing these tasks (Strauss, 2006). Additionally, Rapid Visual Input Processing Tasks (RVIP) and Continuous Performance Task (CPT), like Serial Subtract 3 and 7, measures task vigilance; however, they measure task vigilance through discrimination events, which is distinct from task vigilance in arithmetic tasks such as serial subtraction 3 and 7 (Sarter, 2001). In the case of RVIP and CPT, the reaction time is a measure of mental energy (Strauss, 2006). Additionally, we used serial subtract 3 and 7 because the two tasks have two different hierarchical cognitive function, with serial subtract 3 requiring psychomotor performance and serial subtract 7 requiring working memory and executive function (Bristow, 2016, Scholey, 2010). Like the serial subtraction tasks, the CPT and RVIP have their own hierarchical structures with CPT being considered a lower level task that measures psychomotor performance as it relates to discrimination while the RVIP is considered a higher-level task that measures the use of working memory and executive function as it relates to task discrimination (Badre, 2008). Additionally, the RVIP is broken down into 3 tasks, two of which (the identification of odd and even numbers) are considered higher level tasks that require working memory and executive function and the discrimination of a single event (tertiary task) which is considered a measurement of a psychomotor task within a task requiring working memory and executive function (Strauss, 2006, Badre, 2008). Boolani and colleagues (2019) in their study examining the association between these measures and gait have reported that each measure is distinctly associated with performance on a distinct motor task.
With that in mind, to our knowledge this is the first study of its kind that measures the difference between synthetic caffeine and natural caffeine infused with adaptogens. Therefore, we felt that we should measure as many distinct aspects of cognition as we could. We did this so that we may be able to elucidate which aspects of cognition may be influenced by the adaptogen infused beverage. Therefore, by finding that there is a significant difference for serial subtract 3 attempts we can state that adaptogen infused natural caffeine influences mental energy on lower level mathematical tasks.
Citations
- Scholey AB, French SJ, Morris PJ, Kennedy DO, Milne AL, Haskell CF. Consumption of cocoa flavanols results in acute improvements in mood and cognitive performance during sustained mental effort. Journal of Psychopharmacology. 2010;24(10):1505–14.
- Boolani A, Lindheimer JB, Loy BD, Crozier S, O’Connor PJ. Acute effects of brewed cocoa consumption on attention, motivation to perform cognitive work and feelings of anxiety, energy and fatigue: a randomized, placebo-controlled crossover experiment. BMC Nutrition. 2017;3(1):8
- StraussE,ShermanEM,SpreenO.Acompendiumofneuropsychologicaltests:administration, norms, and commentary. Oxford: American Chemical Society; 2006.
- Sarter M, Givens B, Bruno JP. The cognitive neuroscience of sustained attention: Where top-down meets bottom-up. Brain Research Reviews. 2001;35(2):146–160.
- Bristow T, Jih CS, Slabich A, Gunn J. Standardization and adult norms for the sequential subtracting tasks of serial 3’s and 7’s. Applied Neuropsychology: Adult. 2016;23(5):372–378.
- Badre D. Cognitive control, hierarchy, and the rostro–caudal organization of the frontal lobes. Trends in Cognitive Sciences. 2008;12(5):193–200.
- Boolani, A., Martin, R., Goodwin, A., Avolio, A., Sur, S., Smith, M. L., & Fulk, G. (2019). Associations for tasks requiring single stimulus and working memory with different aspects of gait and posture: an exploratory study. International Journal of Rehabilitation Research, 42(2), 160-167.
The concern is that there is a finding among many insignificant findings, and that this is perhaps overly emphasized. I would like there to be more discussion of the overall lack of findings, particularly on those assessments that are tests (not questionnaires) and ought to be somewhat more objective measures of specific cognitive variables.
We understand the reviewers’ comment in this context. However, we must highlight the fact that while “seemingly” we observed few statistically significant results among the treatments (using stringent statistical analysis), many of our observations provided results that tended trending the same direction. For most cognitive tasks, when participants consumed synthetic caffeine they reported a spike in performance 30 minutes post consumption, and then tapering off after that. However, when participants consumed the e+shot energy beverage a gradual improvement in cognitive task performance was noted. As noted in the statistical analysis section (lines 375-385) most of our measures violated assumption of homoscedasticity, but with a lack of non-parametric analyses for a repeated measures ANCOVA, we had to run parametric analyses which yielded not significant results. However, if subject size was larger, data was more normally distributed and/or the tests had been conducted for longer (i.e. 2.5-3 hours post consumption), our results might have differed. Per this reviewer’s suggestion below we added a statement in our discussion (lines 528-529 and lines 630-632) to reflect that if the study had been conducted for a longer period of time we may have noted results similar to Srivastava and colleagues (2017).
- Srivastava S, Mennemeier M, Pimple S. Effect of Alpinia galanga on Mental Alertness and Sustained Attention With or Without Caffeine: A Randomized Placebo-Controlled Study. Journal of the American College of Nutrition. 2017;36(8):631–9.
The other main issue is with Table 4 which presents the mean and SD for all the variables of interest over the 3 conditions (placebo, caffeine, e-shot). I applaud the authors for being inclusive of all their data. However, the table itself is overwhelming. Most of the numeric values are meaningless without more context. I would recommend considering options to re-organize the presentation of this information, including demonstrating significance of comparisons. This might be organized into separate tables for self-report, physiologic measures, and direct cognitive assessment. This would be easier for readers to review, and by showing significant comparisons within the table(s) there would be greater clarity and context.
We agree with this reviewer and have modified Table 4 to make it easier to follow and understand.
In light of the lack of significant findings (especially for the direct cognitive assessments), would the authors posit that this was a measurement issue? Or some other explanation for the minimal significant findings? Could they suggest improvements for future research that might improve upon their study and increase the likelihood of meaningful significant findings?
You have brought up some excellent points in this comment. As we acknowledged in our discussion (lines 476-479), to our knowledge this is the first study of its kind to systematically compare the effects of synthetic caffeine to naturally sourced caffeine infused with adaptogens. With this in mind, we measured several aspects of cognition, mood and physiologic function. As noted in the statistical analysis section the data violated homoscedasticity (lines 375-385) and with no viable non-parametric tests we were forced to run parametric tests, which may have results in many of the non-significant findings. We have now reported this in our limitations section (lines 630-632) and also pointed out in our discussion section (line 528-529) that maybe if the study had been conducted for a longer period of time we have noted statistically significant results. We also emphasize that fact in our conclusion as we give the future direction of research (lines 641-643).

Round 2
Reviewer 2 Report
Please check the attached document.

Author Response
TRANSMITTAL LETTER
Nutrients- 836371
“Caffeine-containing adaptogenic-rich drink modulates the effects of caffeine on mental performance and cognitive parameters”
Dear Dr. Miljanovic and Distinguished Reviewers,
Thank you for reviewing our manuscript. We appreciate the thorough reviews and valuable feedback. What follows is a point-by-point description of how we have addressed all the concerns expressed by the reviewers. Our responses are in red text and bullet-points below the original reviewers comments, which is in black text.
About my previous suggestions “I could not find the Figure 1” and “I could not look the total Figure 2ʼs gains”. The authorsʼ responses were “Figure 1 is the CONSORT flow diagram. Although it was initially provided in our original manuscript submission we will ensure that itʼs again attached with our edits” and “Figure 2 is the testing flow diagram. Perhaps the reviewer could clarify this comment as we are unsure what the reviewer is requesting regarding this Figure 2.” I checked the previous and present manuscripts, but I still could not find them. The following is the snap shoot around figure 1 and 2 from your revised manuscript. I am sorry, but if these are final version I cannot evaluate this manuscript.
We have re-formatted the manuscript to ensure that Figure 1 and Figure 2 appear in it.
